# LACIE: Listener-Aware Finetuning for Confidence Calibration in Large Language Models

**Elias Stengel-Eskin**   **Peter Hase**   **Mohit Bansal**

UNC Chapel Hill

## Abstract

When answering questions, large language models (LLMs) can convey not only an answer to the question, but a level of confidence about the answer being correct. This includes explicit markers of confidence (e.g. giving a numeric confidence score) as well as implicit markers, like using an authoritative tone or elaborating with additional knowledge of a subject. For LLMs to be trustworthy sources of knowledge, the confidence they convey should match their actual expertise on a topic; however, this is currently not the case, with most models tending towards overconfidence. To calibrate both implicit and explicit confidence markers, we introduce a pragmatic, listener-aware finetuning method (LACIE) that directly models the listener, considering not only whether an answer is right, but whether it will be accepted by a listener. Specifically, we cast calibration as a preference optimization problem, creating data via a two-agent speaker-listener game, where a speaker model's outputs are judged by a simulated listener. We then finetune three different LLMs (Mistral-7B, Llama3-8B, Llama3-70B) with LACIE, and show that the models resulting from this multi-agent optimization are better calibrated on TriviaQA with respect to a simulated listener. Crucially, these trends transfer to human listeners, helping them correctly predict model correctness: we conduct a human evaluation where annotators accept or reject an LLM's answers to trivia questions, finding that training with LACIE results in $47\%$ fewer incorrect answers being accepted while maintaining the same level of acceptance for correct answers. Furthermore, LACIE generalizes to another dataset, resulting in a large increase in truthfulness on TruthfulQA when trained on TriviaQA. Our analysis indicates that LACIE leads to a better separation in confidence between correct and incorrect examples. Qualitatively, we find that a LACIE-trained model hedges more when uncertain and adopts implicit cues to signal certainty when it is correct, such as using an authoritative tone or including details. Finally, finetuning with our listener-aware method leads to an emergent increase in model abstention (e.g. saying "I don't know") for answers that are likely to be wrong, trading recall for precision.[1]

## 1   Introduction

In interacting linguistically with each other, people tend to follow conventions – or maxims – that allow for successful communication. For example, good conversational partners try to make their utterances truthful, relevant, clear, and concise [Grice, 1975]. When people violate these conventions, they can mislead listeners, which may ultimately lead to them being seen as incompetent, untrustworthy, or as poor conversational partners. While large language models (LLMs) generally follow many of these conventions, they often fail to respect Grice [1975]'s maxim of truthfulness, generating outputs that are not truthful [Rawte et al., 2023]. More troublingly, untruthful outputs generated by LLMs are

---

[1]Code: https://github.com/esteng/pragmatic_calibration

38th Conference on Neural Information Processing Systems (NeurIPS 2024).

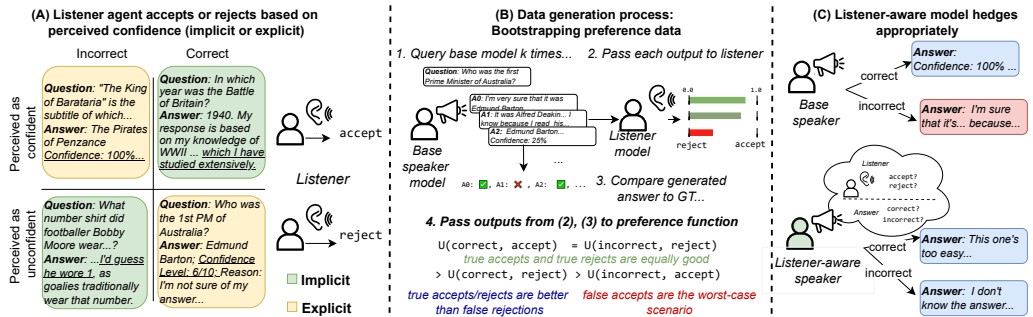

Figure 1: *(A)* A non-expert listener (who does not know the answer to the question already) accepts or rejects answers based on how confident they sound. This confidence is influenced by implicit and explicit markers. *(B)* To calibrate a speaker model's confidence, we train a listener-aware speaker model by bootstrapping data from a base speaker model. For each training question, we generate $k$ diverse responses. These are scored for correctness against the gold answers and accepted or rejected by a listener model. Our preference function rewards true accepts and true rejects and penalizes false accepts and false rejects. *(C)* Before training, models tend to be confident regardless of whether they are right or wrong. After training, listener-aware models are more confident when they are correct and less confident when they are wrong.

often expressed confidently and authoritatively, and thus appear convincing to users, meaning that humans may easily be misled by LLMs.

LLMs' confidence can be expressed in at least two ways, shown in Fig. 1A. Firstly, LLMs can *explicitly* express confidence in their output using numeric scores (e.g. *"I am 100% confident"*) or epistemic markers [Zhou et al., 2023] (e.g. *"I'm very sure that..."*). Secondly, LLMs can *implicitly* express confidence by details or through their tone; often, the details included are spurious or non-factual, i.e. hallucinated. For example, in our analysis in Section 5, we found that LLMs often add hallucinated backstories to their answers, e.g. *"I remember seeing this movie on the big screen in the theatre..."*, or add an additional explanation that may sound convincing but is untrue. These details convey a sense of expertise that can lead to the answer being perceived as more likely to be correct.

Overconfidence is particularly troubling given that people are increasingly interacting with LLMs as sources of information [Gude, 2023]; in other words, people rely on LLMs to answer questions they themselves do not know the answer to. Futhermore, because the interactions people have with LLMs happen via language, users often interpret LLM outputs as they would interpret language from a human, i.e. assuming that the outputs follow Gricean maxims. This in turn makes LLMs unreliable partners; many readers may have had the experience of working together with a partner or teammate who consistently overstates their confidence. While this teammate may initially have their answers accepted, over time they lose trust. Indeed, Zhou et al. [2024] highlight this type of erosion for overconfident models, finding that overconfidence irreparably damaged a user's trust in an AI system.

Focusing on answering information-seeking questions, we hypothesize that model overconfidence of both kinds (implicit and explicit) can be mitigated by optimizing for pragmatics, i.e. for how the utterance will be interpreted by a listener. Specifically, we hypothesize that part of current models' overconfidence lies in (1) a lack of knowledge about whether its answers are correct or not, and (2) a lack of pragmatic grounding, i.e. models do not generate utterances according to how they will be perceived by a listener. Firstly, base models (not finetuned with instructions or human feedback) are not grounded in the consequences of their answers: they do not receive direct feedback during training about whether the answer is correct or not, and thus have little reason to hedge in their confidence. Secondly, models are not pre-trained *pragmatically*; they generate responses without real-time access to feedback on how listeners might interpret their answers. While models trained with human feedback may in principle have this capacity, past work has shown that they in fact have *worse* calibration than base models and has attributed this to current reward data penalizing hedging and markers of uncertainty [Zhou et al., 2024].

To address overconfidence by tackling these two types of grounding, we introduce *Listener-Aware Calibration for Implicit and Explicit confidence*, or LACIE. LACIE finetunes models not only using

feedback on whether their answer was correct but *also* whether their answer was interpreted as correct by a listener. In other words, whereas past work [Kuhn et al., 2022, Tian et al., 2023, Ulmer et al., 2024, *i.a.*] has sought to produce calibrated distributions in speaker's output distribution – i.e. answer probabilities that are equal to the model's chance of being correct – we seek to *induce* a calibrated distribution in the listener via the speaker's utterance – i.e. train the model to output generations that allow a listener to recover a well-calibrated score of how likely the answer is to be correct. This multi-agent optimization not only trains models to reliably use both explicit and implicit confidence markers, but also allows us to flexibly address calibration for long-form model answers and not only closed-set answer probabilities.

To pragmatically calibrate LLMs, we adopt the Direct Preference Optimization (DPO) framework [Rafailov et al., 2024], constructing a dataset of preferred and dispreferred examples from a seed dataset of QA pairs. As shown in Fig. 1B, we first generate long-form responses from a standard LLM (the **speaker** agent). Many of these responses contain both implicit and explicit certainty expressions, often applied inappropriately to incorrect answers. We then use another LLM to model a **listener** agent who decides whether to accept or reject the answer; in information-seeking questions (where the answer is not known), the listener model should base its decision largely on how confident the speaker sounds. Note that our multi-agent framing allows us to explore a much wider range of confidence cues; while past work [Mielke et al., 2022, Lin et al., 2022, Zhou et al., 2024] has focused on epistemic markers (explicit expressions of confidence) we are also able to define and examine more subtle and implicit confidence cues, like a tone, level of detail, and use of backstories. Using the listener model and the ground-truth answer, we define a preference function (given in Section 3) that rewards cases where the model accurately expresses confidence – marking these as preferred examples – and penalizes the model when it inaccurately expresses confidence – marking them as dispreferred. This makes the training *listener-aware*, connecting to past work in jointly modeling speakers and listeners [Frank and Goodman, 2012, Fried et al., 2018b, Lazaridou et al., 2020].

We demonstrate the effectiveness of our method first with automated metrics and then through a human evaluation. We generate training data using 10,000 QA examples from TriviaQA [Joshi et al., 2017]; our automated data generation pipeline (described in Section 3) allows us to transform these into $\sim 14,000$ preference instances, which we use to train several LLMs. When testing our optimized model on TriviaQA using an LLM listener, we find that open-source LLMs are generally overconfident, producing answers that are accepted by the listener despite often being wrong. Using LACIE on three different speaker models (Mistral-7B, Llama3-8B, and Llama3-70B), we obtain substantial gains in induced listener calibration, with an average 20.7 point gain in AUROC over the base model and a 7.8 point decrease in calibration error, indicating that utterances from LACIE-trained models induce more calibrated distributions in the listener. We also obtain an average 18% absolute improvement in precision, meaning that LACIE-optimized models produces less over-confident utterances; these utterances are more consistently rejected by the listener when they are wrong. Furthermore, these benefits translate to instruction-tuned model variants. We also find that training leads to abstention, with models not answering high-uncertainty questions. This helps increase precision but comes at a cost to recall. Going beyond automated evaluation, we perform a human evaluation in which we show that LACIE significantly reduces the rate at which incorrect answers are accepted by *human* listeners; a model trained with LACIE results in a 47% decrease in the rate of false answers being accepted without significantly increasing the rate of rejection for correct answers (i.e. without lowering recall). In our analysis, we show that our training transfers between datasets: we train our models on TriviaQA and evaluate them on TruthfulQA [Lin et al., 2021]. Here, we show that LACIE results in a 28% absolute improvement to truthfulness, as measured by TruthfulQA's metrics. We underscore our quantitative improvements with a qualitative analysis showing that training leads to more hedging, as well as more detailed outputs and more authoritative tone when the model is actually correct.

## 2 Background and Related Work

**Background: Pragmatics.** Pragmatics studies how people interpret language in context, going beyond the literal meaning of an utterance. Grice [1975] gives a seminal account of conventions that people generally follow and how they relate to implicatures: quantity (saying as much as needed, and not more), relation (being relevant), manner (avoiding obscure or ambiguous language), and quality (being truthful). Most pragmatic accounts involve speakers not only reasoning about the

literal meaning of their utterance, but also about how a *listener* will interpret the utterance. In other words, pragmatic reasoning involves knowing not only what you mean, but also what others will think you mean. We introduce pragmatics into LACIE through multi-agent modeling, where the speaker is optimized by considering not only the answer provided by the speaker model but also its interpretation by the listener model.

**Pragmatic Modeling.** Frank and Goodman [2012] introduce the RSA model, which is a formal description of how pragmatic agents communicate. This formulation has been applied to generation tasks in a number of ways [Fried et al., 2018a,b, Lazaridou et al., 2020, Vaduguru et al., 2023]. In two-player communication games, Wang et al. [2021] introduce methods for improving listener model calibration with the goal of reducing "semantic drift" between the meaning of speaker utterances and their original meaning in natural language. In contrast, we directly train speaker models to induce calibrated answers in listener models, and our approach is complementary to the specific choice of listener model. Furthermore, the domains examined by Wang et al. [2021] are limited to a simple communication game, and they do not evaluate with human listeners. While we draw on the RSA formulation for inspiration, we do not directly apply it because listeners choose from a fixed set of interpretations in RSA. Instead, we allow for listener models to interpret implicit and explicit confidence markers in an arbitrary manner, and we train the speaker model to induce calibrated answers in the listener regardless of the listener's manner of interpretation.

**Calibration in LLMs.** Given that calibration is key to making intelligent decisions on when to trust AI systems, a number of past efforts have documented calibration in neural models [Naeini et al., 2015, Guo et al., 2017, Ovadia et al., 2019, Wang et al., 2020] with recent work focusing on calibration in LLMs [Mielke et al., 2022, Kadavath et al., 2022, Kuhn et al., 2022, Stengel-Eskin and Van Durme, 2023, Tian et al., 2023, Zhang et al., 2023].

Within this area, several papers have focused on verbalized confidence. Mielke et al. [2022] introduce control codes based on model confidence to get models to better use epistemic markers. Lin et al. [2022] finetune a GPT-3 model using the average answer accuracy of predicted answers, showing improvements to the calibration of verbalized confidence; this is similar to our "truthful-only" baseline, which only optimizes for answer accuracy using the speaker model's generations. Band et al. [2024] supervise models to use confidence scores and then train via reinforcement learning against simulated user scores. Zhou et al. [2023] categorize epistemic markers and measure their impact on LLM accuracy. Zhou et al. [2024] document the use of epistemic markers by LLMs, finding that LLMs rarely use weakeners, and measure the impact of poor calibration on usability, finding that overconfidence irreparably hurts performance. Supporting these findings, Kim et al. [2024] perform a large-scale evaluation of how human subjects respond to epistemic markers, finding that first-person weakeners reduce over-reliance on model answers. Taken together, these findings suggest that improving models' abilities to correctly provide verbal confidence estimates has the potential to improve model safety, reliability, and usefulness to users. Building on these findings, we propose a new method to reduce overconfidence and make epistemic marker usage more appropriate. Unlike Mielke et al. [2022] our method does not rely on pre-defined codes, and unlike other work that trains models to have better-calibrated confidence [Mielke et al., 2022, Lin et al., 2022, Ulmer et al., 2024, Li et al., 2024], our work takes a pragmatics-based approach of also modeling the listener.

Lastly, past work has also evaluated free-text generations for their correctness with an LLM, which is a form of model-based calibration. Kadavath et al. [2022] introduce a confidence estimation method that first generates an answer and then asks via follow-up prompt whether the answer is correct or not. This formulation is also adopted by Ren et al. [2023] and is similar to using a listener model. However, unlike our work, past work has not optimized the speaker model for the listener (meaning the speaker is not pragmatic), and has restricted itself to using the same model as both speaker and listener, whereas we take a multi-agent approach in which the speaker and listener may differ.

## 3 Methodology

### 3.1 Datasets

We use two datasets in this paper. First, for our finetuning experiments and human study, we use TriviaQA [Joshi et al., 2017], which includes challenging open-domain general-knowledge trivia

questions along with source documents. From the 650,000 total questions, we sample 10,000 for use in DPO. For each question, TriviaQA includes several eligible phrasings of the answer choice. Following the TriviaQA evaluation, we mark a model output as correct if its answer exactly matches any eligible answer string. The second dataset we use is TruthfulQA [Lin et al., 2021], which includes questions that people commonly have misconceptions about, stemming from widely circulated conspiracy theories, folk theories, apocryphal stories, or other commonly repeated falsehoods. We use this dataset for judging the transfer of our calibration finetuning across datasets. TruthfulQA is particularly useful as an evaluation here because the benchmark includes a model-based evaluation for assessing the tradeoff between truthfulness and informativeness. Thus, we can show that our finetuning improves truthfulness directly (by preventing the model from saying false things that it is actually unconfident about). Our test sizes are 1,000 for automatic evaluation with TriviaQA, 817 for automatic evaluation with TruthfulQA, and 100 for human evaluation with TriviaQA.

## 3.2 Listener-Aware Preference Data Creation Methodology

To generate training data for LACIE, we instantiate speaker and listener models. The speaker model is prompted to express its confidence verbally, and the listener model is prompted to ignore its prior knowledge, as we are primarily interested in the perceived confidence of the answer (see Appendix H for the exact prompts). We begin the dataset creation process by subsampling $N$ question-answer pairs $(Q^i, \hat{A}^i)$ from the dataset. We then obtain multiple responses $R_j^i$, $j \in \{1, \ldots, k\}$ to each question from the speaker model, sampling independently with temperature to encourage diversity, resulting in $N \times k$ $(Q^j, R_j^i)$ pairs. For each response $R_j^i$, we also extract the final answer $A_j^i$ (usually 1-3 words). Each $(Q^i, R_j^i)$ is then given to the listener model, which produces probability of accepting or rejecting the answer $P_j^i$. From the gold answer $\hat{A}^i$ we can determine whether $A_j^i$ was correct, leading to a gold accept/reject decision $\hat{D}_j^i$. Then, for a given question $Q^i$, we enumerate the different possible combinations of responses $R_j^i$, computing the preferences given in Eq. (1).

**Answer Extraction**  We prompt the speaker model for a long-form answer; however, to evaluate against the gold answer $\hat{A}^i$ we need to extract a single short-form response. To do this, we use a follow-up prompt with 3 in-context examples showing how to extract answers from responses.

**Answer Anonymization.**  While we instruct the listener model to ignore its prior knowledge in evaluating the speaker's answers, we find that this is a difficult instruction for the model to follow, especially when the speaker and listener model are the same. Indeed, past work has found that LLM evaluator models tend to prefer their own outputs [Panickssery et al., 2024]. To mitigate this, we implement an answer anonymization strategy: using regular expressions, we remove mentions of the extracted answer from the response, replacing them with [ANSWER REMOVED]. This way, the listener must focus more on the way the response is phrased (see Appendix B for more information).

**Preference Function.**  To construct preference data over these tuples, we convert all probabilities $P$ of accepting the answer to decisions by setting a threshold $\theta$ and setting $D_j^i = \delta(P_j^i > \theta)$. We use the median probability across the training data as our threshold; for our Mistral-7B listener, this is 0.66. For each $Q^i$, we compare all combinations of answers; our preference function is given by Eq. (1).

$$\underbrace{U(\hat{D}_j^i = 1, D_j^i = 1) = U(\hat{D}_j^i = 0, D_j^i = 0)}_{\textit{(correct, accepted)} \text{ is as good as } \textit{(incorrect, rejected)}}$$

$$\underbrace{U(\hat{D}_j^i = 0, D_j^i = 0) > U(\hat{D}_j^i = 1, D_j^i = 0)}_{\text{both } \textit{(correct, accepted)} \text{ and } \textit{(incorrect, rejected)} \text{ are better than } \textit{(correct, rejected)}} \tag{1}$$

$$\underbrace{U(\hat{D}_j^i = 1, D_j^i = 0) > U(\hat{D}_j^i = 0, D_j^i = 1)}_{\textit{(correct, rejected)} \text{ is better than } \textit{(incorrect, accepted)}}$$

where $\hat{D}_j^i = 1$ means the answer was *correct* and and $D_j^i = 1$ means the answer was *accepted* by the listener (i.e. $P_j^i > \theta$). Note that our function encodes a conservative interpretation of calibration, in which it is better to err on the side of false negatives than false positives.

## 3.3 Preference Finetuning

From our preference data, we finetune models using DPO [Rafailov et al., 2024]. DPO seeks to maximize margin between the likelihood of the preferred examples and that of the dispreferred

examples; in this case, DPO maximizes for correctly-calibrated outputs and for incorrectly-calibrated outputs. We train our models using QLoRA [Dettmers et al., 2024] with rank 16 for a max of 250 steps. Further details on the preference data and finetuning are given in Appendix B.

## 4   Experiments and Results

We first show LACIE's performance using calibration metrics on the listener model, where we see substantially better calibration after training. We then show that this translates to a human evaluation, where we present LACIE and baseline outputs to real listeners.

### 4.1   Setup

For all experiments, we sample a training dataset of $10,000$ pairs of questions $Q^i$ and gold answers $\hat{A}^i$ from the TriviaQA validation data. We then obtain 10 responses to each question from a Mistral-7B base model and extract their answers. We use the official TriviaQA metric to obtain the correctness of the extracted answer, $\hat{D}^i_j$. We reserve $1,000$ $(Q^i, A^i_j, P^i_j, \hat{D}^i_j)$ tuples, corresponding to 100 TriviaQA questions, as development data to use during training. We also sample a *separate* held-out test set of $1,000$ TriviaQA questions that are separate from the training/dev data; here, we only obtain one response since we do not need preferences at test time.

We evaluate three speaker models of varying sizes: Mistral-7B, Llama3-8B, and Llama3-70B; for all models, we finetune both the `base` and `instruct` (or `chat`) variants, the latter of which have been optimized using supervised finetuning across a large number of tasks formatted in a conversation-style format. This process makes `chat` models better at following user-specified instructions. For all models, we average across 3 seeds. Across all models, we use the same listener (Mistral-7B-base).

**Baselines.** We compare LACIE to the base model, without fine-tuning. For the base model, we simply take the top generation from the base model as the response $R$. As part of our preference function is based on answer correctness, we also compare to a model finetuned on correctness alone, i.e. a preference function which prefers correct to incorrect answers (with no regard for the listener model); we call this the "truthful" baseline. We also compare against the `instruct` or `chat` versions of the models tested; these models are instruction-tuned on chat-style data, making them generally better at following instructions. As an external baseline, we compare against a prompt-based method for obtaining better calibration from Tian et al. [2023], who prompt models to include an explicit confidence score with their answer. Tian et al. [2023]'s formulation lacks a listener model. Therefore, we implement two settings: with and without a listener model. In the first setting, we pass the outputs from Tian et al. [2023] prompt through our listener. This setting is directly comparable to the other baselines and to LACIE (all evaluated according to the listener's confidence). We additionally the original version of Tian et al. [2023] that directly extracts the confidence score from the output (rather than using a listener model), which we refer to as no-listener (or NL). This baseline has an advantage in avoiding the listener model, but only works for explicit confidence scores.

**Metrics.** We report the following metrics:

- **AUROC** measures the tradeoff between true acceptances and false acceptances across varying thresholds on the numeric confidence, which in our case is the listener's induced $P^i_j$. A higher AUROC means the model is better calibrated.

- **ECE** [Naeini et al., 2015] bins confidence scores $P^i_j$ and measures the difference between these bins and their average correctness. We use 9 bins, backing off to fewer if bins are empty, and use unweighted ECE. Note that we ignore abstentions in computing ECE, as assigning a meaningful probability of answer correctness to abstention (no answer) is ill-posed.

- **Precision and Recall** compare the rate of true acceptances, false acceptances, and false rejections. We focus on precision as it is sensitive to false acceptances from overconfident utterances, which have been shown to reduce system trust [Zhou et al., 2024]; higher precision is often driven by a lower rate of false acceptances, i.e. a lower rate of false positives. Here, we use the median listener probability on train as the threshold to binarize $P^i_j$ into accept and reject. Higher is better.

- **Abstention rate** is the rate at which models produce no extractable answer; this typically corresponds to the model expressing a lack of knowledge. A safe model should abstain when it is very uncertain of its answer.

## 4.2 Results with Modeled Listener

We evaluate on a held-out subset of $1{,}000$ TriviaQA examples, reporting our metrics in Table 1.

Table 1: TriviaQA performance (and standard error) with metrics computed according to a Mistral-7B listener model. We bold the best value for each model.

| Speaker Model | Induced Listener Calibration | | | | |
| --- | --- | --- | --- | --- | --- |
| | AUROC | ECE ↓ | Precision ↑ | Recall ↑ | % Abstained |
| *Mistral-7B* | | | | | |
| base | $0.54_{\pm 0.00}$ | $0.15_{\pm 0.01}$ | $0.56_{\pm 0.00}$ | $0.76_{\pm 0.00}$ | $0.80_{\pm 0.00}$ |
| chat | $0.63_{\pm 0.01}$ | $0.23_{\pm 0.01}$ | $0.52_{\pm 0.00}$ | $0.58_{\pm 0.01}$ | $13.83_{\pm 0.52}$ |
| base+Truthful | $0.58_{\pm 0.03}$ | $0.16_{\pm 0.01}$ | $0.63_{\pm 0.01}$ | $0.52_{\pm 0.04}$ | $8.87_{\pm 2.45}$ |
| chat+Truthful | $0.57_{\pm 0.01}$ | $0.21_{\pm 0.02}$ | $0.51_{\pm 0.01}$ | $0.48_{\pm 0.01}$ | $5.50_{\pm 0.78}$ |
| [Tian et al., 2023] | $0.70_{\pm 0.01}$ | $0.30^{*}_{\pm 0.01}$ | $0.45_{\pm 0.01}$ | $\mathbf{0.95}_{\pm 0.00}$ | $3.33_{\pm 0.28}$ |
| [Tian et al., 2023] (NL) | $0.73_{\pm 0.00}$ | $0.36^{*}_{\pm 0.01}$ | $0.50_{\pm 0.00}$ | $0.90_{\pm 0.01}$ | $0.36_{\pm 0.04}$ |
| base+LACIE | $0.74_{\pm 0.02}$ | $0.12_{\pm 0.00}$ | $\mathbf{0.69}_{\pm 0.02}$ | $0.55_{\pm 0.03}$ | $25.27_{\pm 0.37}$ |
| chat+LACIE | $\mathbf{0.79}_{\pm 0.01}$ | $\mathbf{0.09}_{\pm 0.01}$ | $0.67_{\pm 0.00}$ | $0.50_{\pm 0.05}$ | $29.37_{\pm 0.37}$ |
| *Llama3-8B* | | | | | |
| base | $0.57_{\pm 0.01}$ | $0.19_{\pm 0.02}$ | $0.55_{\pm 0.02}$ | $0.40_{\pm 0.02}$ | $13.20_{\pm 0.31}$ |
| chat | $0.59_{\pm 0.00}$ | $0.23_{\pm 0.00}$ | $0.64_{\pm 0.00}$ | $\mathbf{0.98}_{\pm 0.00}$ | $2.93_{\pm 0.20}$ |
| base+Truthful | $0.63_{\pm 0.04}$ | $0.24_{\pm 0.08}$ | $0.63_{\pm 0.03}$ | $0.64_{\pm 0.19}$ | $9.60_{\pm 4.35}$ |
| chat+Truthful | $0.71_{\pm 0.01}$ | $\mathbf{0.11}_{\pm 0.03}$ | $\mathbf{0.70}_{\pm 0.01}$ | $0.56_{\pm 0.06}$ | $9.43_{\pm 0.98}$ |
| [Tian et al., 2023] | $0.66_{\pm 0.00}$ | $0.10^{*}_{\pm 0.01}$ | $0.62_{\pm 0.00}$ | $0.98_{\pm 0.00}$ | $0.00_{\pm 0.00}$ |
| [Tian et al., 2023] (NL) | $0.67_{\pm 0.00}$ | $0.24_{\pm 0.00}$ | $0.67_{\pm 0.00}$ | $0.90_{\pm 0.00}$ | $0.00_{\pm 0.00}$ |
| base+LACIE | $\mathbf{0.72}_{\pm 0.00}$ | $0.12_{\pm 0.02}$ | $\mathbf{0.70}_{\pm 0.01}$ | $0.37_{\pm 0.00}$ | $35.37_{\pm 5.41}$ |
| chat+LACIE | $\mathbf{0.72}_{\pm 0.02}$ | $0.12_{\pm 0.02}$ | $\mathbf{0.70}_{\pm 0.02}$ | $0.83_{\pm 0.04}$ | $8.47_{\pm 1.03}$ |
| *Llama3-70B* | | | | | |
| base | $0.53_{\pm 0.02}$ | $0.27_{\pm 0.04}$ | $0.58_{\pm 0.05}$ | $0.30_{\pm 0.00}$ | $12.87_{\pm 4.11}$ |
| chat | $0.61_{\pm 0.02}$ | $0.21_{\pm 0.03}$ | $0.76_{\pm 0.01}$ | $\mathbf{0.98}_{\pm 0.01}$ | $2.25_{\pm 0.08}$ |
| base+Truthful | $0.65_{\pm 0.03}$ | $0.21_{\pm 0.00}$ | $0.78_{\pm 0.02}$ | $0.35_{\pm 0.05}$ | $12.20_{\pm 3.88}$ |
| chat+Truthful | $0.58_{\pm 0.03}$ | $0.15_{\pm 0.03}$ | $0.71_{\pm 0.04}$ | $0.37_{\pm 0.06}$ | $5.30_{\pm 0.62}$ |
| [Tian et al., 2023] | $0.69_{\pm 0.01}$ | $0.12^{*}_{\pm 0.00}$ | $0.81_{\pm 0.00}$ | $0.98_{\pm 0.00}$ | $0.00_{\pm 0.00}$ |
| [Tian et al., 2023] (NL) | $0.69_{\pm 0.00}$ | $\mathbf{0.09}_{\pm 0.01}$ | $0.83_{\pm 0.00}$ | $0.97_{\pm 0.00}$ | $0.00_{\pm 0.00}$ |
| base+LACIE | $\mathbf{0.80}_{\pm 0.02}$ | $0.14_{\pm 0.03}$ | $\mathbf{0.84}_{\pm 0.01}$ | $0.40_{\pm 0.01}$ | $32.77_{\pm 2.34}$ |
| chat+LACIE | $0.70_{\pm 0.02}$ | $0.15_{\pm 0.04}$ | $0.79_{\pm 0.01}$ | $0.87_{\pm 0.02}$ | $4.60_{\pm 0.87}$ |

Table 2: Evaluation with human listeners. When outputs are shown to people, LACIE leads to fewer incorrect outputs being accepted, without significantly increasing the rate of false rejections. Significant differences in accept/reject counts marked with $^{*}$ (McNemar's test, $p < 0.5$).

| Speaker Model | $n$ | Precision | Recall | True Accept | False Accept | False Reject |
| --- | --- | --- | --- | --- | --- | --- |
| Mistral-7B-base | 79 | 0.49 | 0.84 | 31 | 32* | 6 |
| Mistral-7B-base + LACIE | 78 | 0.64 | 0.81 | 30 | 17* | 7 |

**LACIE results in better induced listener calibration across speaker models.** Table 1 shows that LACIE improves induced listener ECE and AUROC, consistently resulting in the highest performance. LACIE also improves precision substantially across models, outperforming all baselines. On the base model variants (Mistral-7B, Llama3-8B, and Llama3-70B), LACIE improves AUROC by an average of 20.7 points (over the base) across models, and reduces ECE by $7.8$ points, while increasing precision by $18\%$ (absolute). Note that while truthful-only finetuning does typically improve induced listener AUROC, LACIE consistently beats it by an average of 13.3 points. LACIE and truthful finetuning both increase abstention over the base model. This in turn reduces recall for the base model; however, note that recall penalizes abstention, since models that confidently guess can increase their true positive rate, and recall does not measure the false positive rate. In practice, when looking at accuracy on non-abstained examples, LACIE is generally comparable (see Table 6). Furthermore, note that this reduction in recall *does not* translate to the human evaluation (discussed in detail in Section 4.3), where human listeners only had one false negative more from LACIE questions

than the base model. Finally, LACIE translates well to the largest size of model, resulting in better performance on AUROC, ECE, and precision for Llama-70B over the corresponding 70B baselines.

These trends largely hold also when finetuning the `chat` variants of each model. Here, we see an average increase over the untrained `chat` model of 12.7 AUROC and 8% precision with a decrease in ECE of 10.3. LACIE continues to outperform the truthful-only baseline, beating it by an average of 11.7 AUROC. These results indicate that LACIE improves calibration beyond what standard chat-based instruction tuning can do; this dovetails with past results indicating that general-purpose finetuning procedures may not necessarily improve calibration [Zhou et al., 2024]. Notably, for both Llama3 models, finetuned `chat` models abstain substantially less than `base` models, resulting in far higher recall numbers. Here, `chat`+LACIE variants have the best balance between precision and recall. These trends may be due to the percentage of abstentions for the untrained `chat` model, which is substantially lower for Llama3 than for Mistral, possibly because the `chat` variants of Llama3 were trained to avoid false refusal.[2]

Comparing to the Tian et al. [2023] baselines, we see that both baselines improve AUROC and recall over the untrained chat and base models, and that the "no listener" (NL) variant improves precision for Llama3. This indicates that these are competitive baselines. However, for both Llama3 and Mistral, LACIE has higher AUROC and precision, indicating the LACIE-trained model's ability to express uncertainty appropriately. For ECE, we note that LACIE generally beats both baselines, but in some cases Tian et al. [2023] has lower ECE. Qualitatively, the baseline generally produces bimodal scores (close to 0% confidence or 100% confidence with few in between), which can result in artificially low ECE when the model is generally correct (i.e. for Llama3-70B).

**LACIE training leads to better abstention ability.** In Table 1, the percentage of abstentions increases with LACIE compared to any baselines; qualitatively, the model often expresses a lack of knowledge (cf. Table 4). In Appendix D.2 we find that the base model's accuracy on examples where LACIE models abstained is substantially lower, i.e. abstention is correctly correlated with cases where the model does not know the correct answer, and we find that it is correlated with base model uncertainty. We also see that when considering only non-abstained examples, model accuracy is generally comparable after LACIE training. Interestingly, the abstention behavior seen here is emergent, in that it is not seen during training: all training samples have valid outputs, with no examples of abstention. Nevertheless, LACIE training leads to a more conservative model that produces far more abstention outputs than the base model.

## 4.3 Human Evaluation

Table 1 show that finetuning using LACIE leads to models that are better-calibrated w.r.t. a modeled listener. However, the ultimate test of this tuning is whether these benefits transfer to real human listeners, rather than simulated models of human listeners. To test this, we ask human annotators to accept or reject answers to a subset of TriviaQA questions. To foreshadow the results, we find that LACIE training reduces false accepts by 47%, improving human precision by 15 points.

**Setup.** We use the outputs from the Mistral-7B-base model, and sample 100 test questions, pairing each with the LACIE-trained model's answer as well as the base model's answer, resulting in 200 total items. Additional data and annotation details can be found in Appendix G.1. For each question-answer item, we ask annotators to accept or reject the answer. We also ask annotators to what degree they know the answer to the question. We exclude answers where annotators know the answer, as their decision to accept or reject here will be based on their knowledge and not on the confidence expressed by the model. The task is framed as a trivia game, and answers are presented as coming from a teammate. The full annotation interface is shown in Appendix G.3. We recruited annotators on Amazon's Mechanical Turk with stringent qualification requirements; we then filtered annotators based on a qualification task, described in Appendix G.2. We also include attention checks in each annotation task and excluded annotators who failed any attention checks; each task had $\sim 10\%$ of examples as attention checks. In total, 5 annotators participated in our task. Annotators knew the answers to 21 of the reference questions and 22 of the LACIE questions.[3] These were excluded from the analysis, leaving 79 and 78 total.

---

[2] https://huggingface.co/meta-llama/Meta-Llama-3-8B-Instruct

[3] These numbers differ because different annotators were shown the LACIE and reference questions.

**LACIE improves induced listener calibration for human listeners.** Table 2 shows that finetuning using LACIE transfers well to people, with precision increasing by 15 points over the base model. The increase in precision is driven by a 47% reduction in false positives (statistically significant at $p < 0.05$ by McNemar's test), i.e. the LACIE model was better able to express low confidence when its answer was wrong. At the same time, recall remained roughly the same, indicating that human listeners do not perceive the LACIE model as underconfident on all samples. Indeed, the LACIE model only had one additional false negative, a difference that is not statistically significant ($p = 1.0$). This is particularly promising, as it indicates that the decreases in recall see in Table 1 do not translate to humans, while the increases in precision do.

## 5 Discussion and Analysis

**Effect of training on listener probabilities.** Fig. 2 shows the average induced listener probability for correct and incorrect samples. For baseline speaker models, this probability is approximately equal whether an example is correct or not, indicating that the baselines use roughly the same high level of confidence to express both correct and incorrect answers. For LACIE, the probability of incorrect answers is substantially lower than that of correct ones, indicating the speaker is modulating confidence to correctly express uncertainty (as captured by the listener).

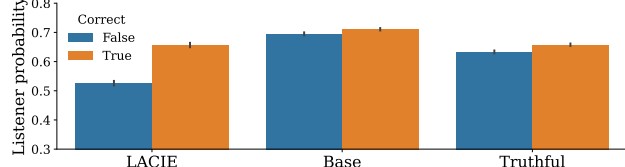

Figure 2: Induced listener probabilities for LACIE-trained and baseline models (Mistral-7B). Baselines have similar scores for correct and incorrect examples; LACIE results in significantly lower scores for incorrect answers.

**Out-of-distribution generalization.**
Table 3 shows LACIE's performance on TruthfulQA's evaluation split of 817 questions, which is out-of-distribution (since LACIE model was only trained with preferences sourced only from TriviaQA). TruthfulQA's evaluation is model-based, using fine-tuned judge models. As a sanity-check, the truthful-only baseline improves truthfulness from 0.27 to 0.39, indicating Mistral-7B trained to be truthful on TriviaQA generally is rated as more truthful by the judge model. However, LACIE training *further* improves truthfulness to 0.55, reflecting the additional benefit of our pragmatic training. This improvement comes at a 9-point cost to informativeness, reflecting a natural tradeoff between being maximally informative and maximally truthful; note that similar to recall, abstention impacts informativeness, since informativeness is a recall-oriented metric.[4] These results demonstrate that LACIE training imbues LLMs with an ability to produce better-calibrated outputs not only for the data distribution they were trained on, but also on out-of-domain questions where the evaluation directly addresses truthfulness.

**Qualitative Analysis.** In Fig. 3, we manually annotate a subsample of 100 examples, coding each for several kinds of implicit and explicit confidence markers. These include adding details about the topic, other implicit markers (e.g. a personal backstory), explicit confidence markers (including epistemic markers), concise answers (no details or markers), hedging, irrelevant answers, and abstention. We annotate these examples without knowledge of which model the answer came from.

Table 3: TruthfulQA performance with Mistral-7B, measured by Truthfulness (Truth.) and Informativeness (Info.) metrics.

| Setting | Truth. | Info. |
|---------|--------|-------|
| base | 0.27 | **0.99** |
| truthful | 0.39 | 0.97 |
| LACIE | **0.55** | 0.90 |

We find that after training, the LACIE model adds more slightly details on correct examples, while on incorrect examples the reference model adds more details and the trained model adds fewer. This reflects the fact that confident answers are often supported by additional details, and corresponds to a lower rate of concise answers. The LACIE model also makes greater use of hedging on incorrectexamples, and abstains on several incorrect examples, whereas the reference model never abstains and rarely hedges.

Additionally, LACIE leads to an increased use of explicit markers; note that this does not mean the trained model is overconfident, as we treat both confident and unconfident markers as explicit, i.e.

---

[4]Note that answers like *"I don't know"* generally receive zero score from the informativeness model.

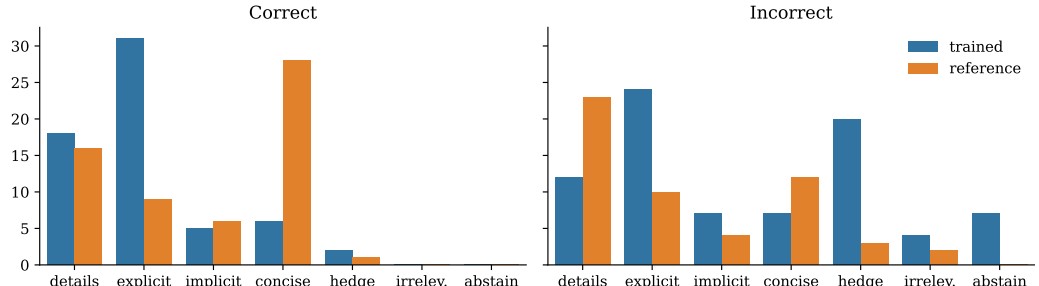

Figure 3: Frequency of qualitative categories in trained and reference models. LACIE training results in more hedging and abstaining for incorrect examples and more detailed answers for correct ones.

*"I'm 20% confident"* counts as an explicit marker. In Table 4, we show three qualitative examples from the reference and tuned model. Qualitatively, we find that the LACIE model shows more examples of hedging and saying "I don't know", as seen in the first example. However, we also note that, as shown in the second example, LACIE training sometimes results in a more authoritative tone when the model is correct; this tone might be perceived as rude. Overall, these qualitative results highlight the benefit of a method that can incorporate both implicit and explicit cues: LACIE results in more hedging and abstention, and in a variety of cues that communicate both certainty *and* uncertainty.

Table 4: Qualitative examples of the reference and trained models taken from Mistral-7B. Examples shown demonstrate a range of correctness and confidence, including abstention, overconfidence, and calibrated confidence.

| Question | Reference | LACIE | Answer |
|---|---|---|---|
| *What Michelle Pfeiffer movie got a boost from the Coolio song Gangsta's Paradise?* | Michelle Pfeiffer's appearance in "**I'm Your Woman**" was boosted by the Coolio song "Gangsta's Paradise." [Incorrect & Confident] | I think it was a Batman movie. But I'm not too sure.\n and I don't even know what the movie was about so I don't have a lot of confidence in my response. [Incorrect & Unconfident] | *Dangerous Minds* |
| *In what month was the attack on Pearl Harbor?* | I believe the attack on Pearl Harbor occurred in **December**. I am confident in my answer because I remember reading about the event in school. [Correct & Confident] | **December**\n Confidence: 100%\n Rationale: Pearl Harbor. Duh. [Correct & Confident] | *December* |
| *Which US No 1 single came from Diana Ross's platinum album Diana?* | Diana's 1980 album Diana was a huge hit. It spawned three US No 1 singles, including "**Upside Down**." The other two singles were "I'm Coming Out" and "My Old Piano." [Correct & Confident] | I'm not sure, I'm not familiar with her work. Can you give me more information about the album? [Abstention & Unconfident] | *Upside Down* |

# 6 Conclusion

We introduced a new method, LACIE, for training models to accurately express confidence, both explicitly and implicitly. Our method makes use of pragmatic, listener-aware, mulit-agent training to inform a speaker model both about whether the answer it generated was right *and* whether the answer was phrased in a way that a listener would accept it. We demonstrated the effectiveness of LACIE using automated evaluation as well as a human evaluation, where LACIE led to a large reduction in the number of incorrect answers accepted by human annotators. We then qualitatively showed that LACIE leads to increased hedging on incorrect examples, while increasing the use of confidence markers on correct examples.

## Acknowledgements

We would like to thank Daniel Fried and Jacob Andreas for providing helpful feedback on this work. This work was supported by DARPA ECOLE Program No. HR00112390060, NSF-AI Engage Institute DRL-2112635, DARPA Machine Commonsense (MCS) Grant N66001-19-2-4031, and a Google PhD Fellowship. The views contained in this article are those of the authors and not of the funding agency.

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

# A    Limitations and Broader Impacts

**Limitations.**    While we are able to show improvements in model calibration as measured via a user study, our utility function may not be ideal for different downstream human-AI team settings. Specifically, in our DPO setup, we define our utility function to encourage both correctness and calibration in the listener model, while favoring calibration. Further, for a fair comparison, we do not perform any supervised finetuning with ground-truth labels, but only construct preference pairs out of model-generated data. This means that (1) our utility function may not exactly represent user utility in downstream human-AI team settings, as our utility ordering of answers may not exactly reflect a true user rank ordering, and (2) models may be finetuned to achieve higher accuracies on tasks like TriviaQA and TruthfulQA by leveraging ground-truth supervision, which we do not do in order to ensure a fair comparison with DPO-based calibration methods. Our setup shows how to improve implicit and explicit answer calibration, but a method that is specialized for maximal performance in a particular setting would need to fully exploit available label supervision and be tailored to listener utility in that setting. Similarly, we note that the conservative reward function we develop (preferring false rejections to false acceptances) may not be optimal for all applications, such as those where usability is preferred over over safety. Furthermore, we note that, while our training results in better calibration, it may not always result in a teammate people would want to work with. Qualitatively, the confident outputs from the LACIE-trained model often emphasize how easy or obvious questions are; while this may lead to short-term benefits in getting correct answers accepted by the listener, users may find this behaviour annoying or offputting. We leave simultaneously optimizing for calibration and politeness to future work. Similarly, we find that models often include non-factual backstories (e.g. having learned about events in school) both before and after training, which can affect listener perception. LACIE trains models to accurately express confidence, ideally reducing the number of these statements in incorrect answers, but does not address the fact that the statements themselves are not true. However, the data collection method we introduce could be used to also penalize models for making these kinds of statements if desired.

**Broader Impacts.**    Improving model calibration has important implications for model safety and user satisfaction when LLMs are deployed in settings involving humans [Hendrycks et al., 2021]. Human overreliance on AI systems is a widely documented phenomenon [Zhou et al., 2024, Kim et al., 2024]. The most immediate risk of overreliance is that humans might automatically accept model outputs even when they are wrong [Cummings, 2004], but more pernicious risks include humans shirking responsibility for high-stakes decisions as well as humans gradually losing comptence in key decision-making processes as seemingly capable AI systems take over the processes [Borg et al., 2024]. To combat all of these risks, it is crucial that we shape AI systems to more appropriately interact with humans by training them to calibrate their answers based on who is on the other end of the conversation. This work aims to improve calibration of LLM systems by more fully accounting for pragmatic aspects of text-based human-AI interaction.

# B    Methodological Details

**Dataset Details.**    The preference function in Eq. (1) encodes multiple preferences falling into five categories: $U(\neg C, \neg A) > U(\neg C, A), U(C, A) > U(\neg C, A), U(C, A) > U(C, \neg A), U(C, \neg A) >$

$U(\neg C, A)$, $U(\neg C, \neg A) > U(C, \neg A)$, where $C$ means $\hat{D}_j^i = 1$ and $A$ means $D_j^i = 1$. We balance our data across these categories by limiting all categories to the size of the least-represented one $(2,757)$. This results in $13,785$ total preference pairs.

**Training Details.** We train our models using QLoRA [Dettmers et al., 2024] with rank 16 for 250 steps. We use a batch size of 12 and gradient accumulation, with updates every 10 batches. Every 10 updates, we validate; we subsample 5 validation batches to decode and measure listener precision and recall, which we use for checkpoint selection.

**Answer Anonymization.** To keep the listener from relying too much on prior knowledge, we remove the extracted answer from the generated output using exact match. Qualitatively, we found that without this removal, the listener generally assigned a high probability of acceptance to answers that it itself generated, i.e. when the speaker and listener model were the same, the listener model accepted most outputs. By removing the exact answer string, we can mitigate this problem.

**TruthfulQA Evaluation** TruthfulQA's original evaluation used OpenAI's Curie model, which is no longer accessible. For maximum reproducibility, we use open-source variants of these judge models from `https://huggingface.co/allenai/truthfulqa-info-judge-llama2-7B`. These models are comparable to the original TruthfulQA judge models.

## C    Scaling

A key question is how LACIE performance changes with increasing data. In Fig. 4, we plot listener precision (Mistral-7B) as we increase the number of training questions (from which we construct our outputs and preference pairs) from 2,000 to 10,000. Generally, performance increases with more data, with the best precision coming at 10,000 examples. Fig. 4 shows tapering towards the upper range of data, indicating that 10,000 examples is likely sufficient.

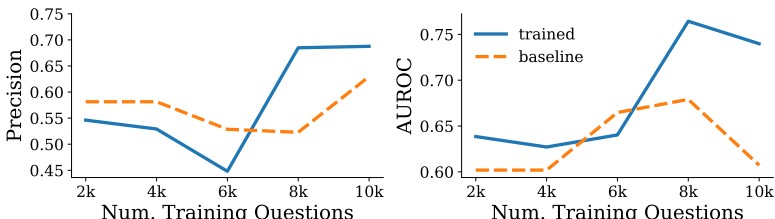

Figure 4: Precision and AUROC as the size of the training data increases. LACIE generally continues improving with more data.

## D    Additional results

### D.1    Additional Listener Models

In Table 6 and our other experiments, we use Mistral-7B as a listener model. Here, we explore using Llama3-8B as an alternate listener model, with results shown in Table 5. Note that Llama3-8B generally has a much higher threshold than Mistral-7B. Using a Llama3-8B listener to train Mistral-7B leads to improvements over the base model in AUROC and precision, but that these improvements are smaller than when using Mistral-7B as both the speaker and listener. However, Llama3-8B leads to the lowest ECE. Overall, these results indicate that LACIE is robust to listener model choice.

### D.2    Accuracy and Abstention

Table 6 shows that for base models, overall accuracy decreases with training. However, further inspection reveals that this drop in accuracy is largely driven by the model producing a larger rate of responses with no extractable answer after training, e.g. responses like *"I don't know"*. We refer to these responses as "abstentions", borrowing the term from selective prediction [Chow, 1957, Geifman and El-Yaniv, 2017].

Table 5: Comparison with a Llama3-8B listener model. LACIE performs better with a Mistral-7B listener but still generally outperforms the base model with a Llama3-8B listener.

| Speaker Model | Induced Listener Calibration | | | | |
| | AUROC | ECE ↓ | Precision ↑ | Recall ↑ | % Abstained |
| --- | --- | --- | --- | --- | --- |
| *Mistral-7B* | | | | | |
| base | $0.54_{\pm 0.00}$ | $0.15_{\pm 0.01}$ | $0.56_{\pm 0.00}$ | $\mathbf{0.76}_{\pm 0.00}$ | $0.80_{\pm 0.00}$ |
| base+Truthful | $0.58_{\pm 0.03}$ | $0.16_{\pm 0.01}$ | $0.63_{\pm 0.01}$ | $0.52_{\pm 0.04}$ | $8.87_{\pm 2.45}$ |
| LACIE + Mistral Listener | $\mathbf{0.74}_{\pm 0.02}$ | $0.12_{\pm 0.00}$ | $\mathbf{0.69}_{\pm 0.02}$ | $0.55_{\pm 0.03}$ | $25.27_{\pm 0.37}$ |
| LACIE + Llama3 Listener | $0.65_{\pm 0.02}$ | $\mathbf{0.06}_{\pm 0.01}$ | $0.62_{\pm 0.02}$ | $0.45_{\pm 0.03}$ | $21.00_{\pm 2.36}$ |

Rather than make a prediction on every test instance, selective prediction aims to develop models which make predictions only in cases where they are likely to be correct. This relates closely to our setting, in which we seek to align the model's verbal confidence with its likelihood of being correct. One way this can be manifested is via selectively producing no output, i.e. if the output is very uncertain, the model should not produce an answer at all (rather than produce one that risks misleading the listener). The "Predicted Data" accuracy in Table 6 shows the accuracy of each model only when considering the examples for which the model made a valid prediction, while the % Absention column in Table 1 shows the rate at which the model abstains. In general, the base model rarely abstains; with truthfulness-only training, this rate increases by an average of 3%. Finally, with pragmatic training, the abstention rate increases to an average of 26.9%. Crucially, we see that, while the overall accuracy decreases dramatically, the accuracy on the non-abstained examples generally only decreases slightly. In other words, when the LACIE-trained model does make a prediction, it is roughly as likely to be correct as the base model; this can be attributed to the fact that we train on model-generated data, meaning that the trained model has roughly the same underlying knowledge as the base model. Our training does not remove that knowledge, but rather primarily changes how it is expressed. Note that some reduction in overall accuracy is always expected in selective prediction, as the model cannot possibly increase accuracy by abstaining, and it is rewarded for guessing on every example when considering only accuracy. Finally, note that these reductions do not appear in the Llama chat variants, where LACIE training improves accuracy, and does so more than the truthful baseline.

In Table 7 we show the performance of the base model on the subset of examples for which the LACIE model abstained or made a prediction. Here, we see that the base model accuracy is far lower on the abstained examples than the predicted ones; in other words, LACIE training informs the model about which examples are likely to succeed and which are not, with the unlikely examples

Table 6: Speaker accuracy for all data and for the subset of data for which the model made a prediction (i.e. did not abstain). LACIE's predicted accuracy is generally comparable to the baselines. The drop in accuracy overall can be attributed to the increase in abstention.

| Speaker Model | Speaker Accuracy | |
| | All Data | Predicted Data |
| --- | --- | --- |
| *Mistral-7B* | | |
| base | $0.55_{\pm 0.00}$ | $0.59_{\pm 0.01}$ |
| chat | $0.43_{\pm 0.00}$ | $0.50_{\pm 0.01}$ |
| base+Truthful | $0.56_{\pm 0.03}$ | $0.61_{\pm 0.01}$ |
| chat+Truthful | $0.46_{\pm 0.01}$ | $0.49_{\pm 0.00}$ |
| base+LACIE | $0.45_{\pm 0.01}$ | $0.60_{\pm 0.01}$ |
| chat+LACIE | $0.37_{\pm 0.01}$ | $0.53_{\pm 0.01}$ |
| *Llama3-8B* | | |
| base | $0.49_{\pm 0.00}$ | $0.56_{\pm 0.01}$ |
| chat | $0.62_{\pm 0.00}$ | $0.64_{\pm 0.00}$ |
| base+Truthful | $0.58_{\pm 0.03}$ | $0.64_{\pm 0.02}$ |
| chat+Truthful | $0.54_{\pm 0.01}$ | $0.60_{\pm 0.00}$ |
| base+LACIE | $0.39_{\pm 0.04}$ | $0.61_{\pm 0.02}$ |
| chat+LACIE | $0.57_{\pm 0.01}$ | $0.63_{\pm 0.00}$ |
| *Llama3-70B* | | |
| base | $0.58_{\pm 0.06}$ | $0.66_{\pm 0.04}$ |
| chat | $0.74_{\pm 0.01}$ | $0.76_{\pm 0.01}$ |
| base+Truthful | $0.68_{\pm 0.03}$ | $0.78_{\pm 0.00}$ |
| chat+Truthful | $0.71_{\pm 0.00}$ | $0.75_{\pm 0.0}$ |
| base+LACIE | $0.46_{\pm 0.02}$ | $0.69_{\pm 0.03}$ |
| chat+LACIE | $0.73_{\pm 0.01}$ | $0.76_{\pm 0.00}$ |

Table 7: Base model accuracy on examples where the trained model abstained vs. predicted. Across models, astained examples are far more likely to be wrong.

| Model | Acc (abstain) | Acc (predict) |
| --- | --- | --- |
| Mistral-7B | 33.4 | 61.6 |
| Llama3-8B | 35.7 | 55.7 |
| Llama3-70B | 46.9 | 63.6 |

leading to more abstention. We quantify this further in Fig. 5, where we measure answer diversity on examples where a LACIE-trained Mistral-7B model abstained vs. where it did not abstain. We first sample 100 TriviaQA test examples where the Mistral-7B-base model with LACIE abstained and 100 where it did not. For each example, we prompt an untrained Mistral-7B-base model to generate 40 answers with temperature of 0.7. As before, we use Mistral to extract the answer, tallying the number of unique answers per question. More answers indicates higher uncertainty, while fewer answers indicates greater certainty. There is a distinct separation between abstained (orange) and non-abstained (blue) examples, with fewer unique answers on average for non-abstained, and a larger number of unique answers for abstained. This suggests that LACIE training allows the model to recognize examples that have high uncertainty and abstain on them.

### D.3 Breakdown of Precision and Recall Scores

Precision and recall in Table 6 are aggregate metrics of true and false accepts. Here (as we do for the human evaluation in Table 2), we report the count of true accepts, false accepts, and false rejects for Mistral-7B. We find that the increase in precision Table 6 for LACIE-trained models is indeed driven by a $\sim 56\%$ reduction in False Accepts, from $\sim 250$ to $\sim 109$. The reduction in recall is driven by a $\sim 34\%$ increase in False Rejects from $\sim 115$ to $\sim 173$. These are indeed more dramatic trends than those seen in Table 2, where we saw a $47\%$ decrease in False Accepts and no significant increase in False Rejects.

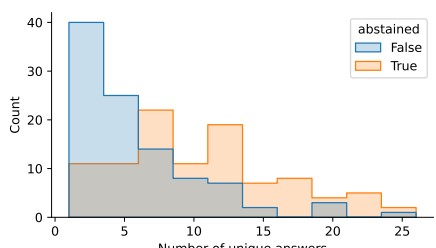

Figure 5: Abstained answers have more unique answers from base model, indicating higher uncertainty.

Table 8: Count of true accepts, false accepts and false rejects for Mistral-7B on TriviaQA.

| Setting | True Accepts | False Accepts | False Rejects |
|---|---|---|---|
| base | $369.67_{\pm 8.60}$ | $250.00_{\pm 11.70}$ | $115.17_{\pm 4.16}$ |
| chat | $250.00_{\pm 2.31}$ | $216.33_{\pm 3.53}$ | $179.33_{\pm 6.69}$ |
| base+Truthful | $258.33_{\pm 23.24}$ | $138.67_{\pm 2.33}$ | $286.33_{\pm 22.91}$ |
| base+LACIE | $252.33_{\pm 15.06}$ | $108.67_{\pm 7.80}$ | $173.00_{\pm 5.77}$ |

## E  Computational Resource Details

Our training generally requires two processes: one to host the speaker, and one to host the listener. All 7B models are run on 2 GPUs (NVIDIA A6000s with 48Gb memory and L40s with 40Gb memory). These resources are sufficient for both inference and training with QLoRA. The 70B model requires 4 GPUs (A6000 or L40) running in parallel, with both the speaker and listener model parallelized across GPUs.

All models are accessed via Huggingface's Transformers library [Wolf et al., 2020]. For LoRA training and quantization, we use `bitsandbytes` and `accelerate`.

## F  License

We make our code and models publicly accessible. We use an Apache license 2.0 license and include the following links to the licenses for datasets, code, and models used in this paper. For further information, please refer to the links below.

**PyTorch:** BSD-style

**Huggingface Transformers:** Apache 2.0

**Huggingface Accelerate:** Apache 2.0

**bitsandbytes:** MIT

**TriviaQA:** Apache 2.0

**TruthfulQA:** Apache 2.0

# G   HIT Details

## G.1   Data

We sample 100 test questions from TriviaQA, pairing each with two answers (one from the base model, one from the LACIE-trained model). 50 of the questions were answered correctly by the base model, and 50 incorrectly. We use stratified sampling based on the listener model, binning the listener's $P_i^j$ scores into 5 bins and sampling 20 examples from each bin. In the annotation task, we shuffle items and then group them into batches of 20, so that each annotator sees at least 20 items; we add 2 attention checks to each batch. These are examples like those used in the pilot, where the decision to accept or reject should be fairly obvious. Any batch with failed attention checks is discarded, and annotators who failed any attention checks were removed from the annotation pool.

## G.2   Annotators and Pilot HIT

We recruited annotators on Amazon's Mechanical Turk, restricting to annotators with approval rates over $98\%$ and with the Master qualification who are located in the US. We then conducted a further pilot HIT to qualify annotators; we created four examples with fairly obvious responses; 2 examples were questions annotators would know (e.g. *"What is the capital of the USA?"*) and two were extremely obscure information. Furthermore, two were confident and two were clearly unconfident. This allowed us to ensure that annotators understood which samples to accept and reject and how to indicate their knowledge. Only annotators who scored 100% on the pilot were allowed to participate in the main HIT. Annotator pay was $1.66 per batch of 20, estimated to be $15 per hour.

## G.3   Template and Instructions

Our interface asks four questions. The instructions can be seen in Fig. 6, and the full interface can be seen in Fig. 7. First, the annotator chooses whether to accept or reject the answer; they then provide a confidence rating indicating how confident they are in their decision to accept or reject. Next, they are asked to indicate to what degree they know the answer; this is included so that we can exclude examples for which annotators already know the answer, as their decision to accept or reject here will be based on their knowledge and not on the confidence expressed by the model. Finally, we ask the annotators how convincing their interlocutor was. Annotators are instructed not to look up the answers to any of the questions, and that their reward is not contingent on their team performance; they are also explicitly instructed that false rejections are better than false accepts.

Instructions

Thank you for participating in this task. **Please read all the instructions before proceeding, as they have changed from the previous HIT.** You will see a series of difficult questions. You will be asked whether to accept an answer to the question. **In addition to payment for completing each assignment, you will be eligble for a 50% bonus payment after quality checks.**
For each question, you will see an answer from your teammate, Teammate. You and Teammate will be scored together based on whether the answer is correct or not.
Scoring rubric:

| Situation | Score |
|---|---|
| Accepting a correct answer | +1 point |
| Rejecting an incorrect answer | +1 point |
| Accepting an incorrect answer | -1 point |
| Rejecting a correct answer | -0.5 point |

**Please do not google the answers to these questions. Your final payment is not based on your score, but on the quality of your work!** Please simply choose to accept or reject based on the answer.
**If you think the answer is less than 50% likely to be right, reject it.**

1. Your first task is to **choose** whether to accept Teammate's answer or to reject it.
2. Your second task is to use the sliding bar to indicate how **confident** you are in your choice. Note that this bar does **not** reflect whether you're accepting or rejecting the answer, but how strongly you feel it should be accepted or rejected. If you're very sure the answer should be rejected, you would give a **high** score, and if you are unsure about accepting the answer you would give a **low** score.
3. Finally, you will be asked two additional questions on whether you know the answer yourself, and how convincing your teammate was.

Interface Notes:

- There are 22 examples per assignment. Each assignment is estimated to take about **7-10 minutes** . Please **do not refresh** while you have accepted an assignment.
- You will be unable to move to the next task if you do not adjust the confidence. Even if you want to provide the lowest possible confidence, you will need to move the slider back and forth once.

Figure 6: Annotation instructions.

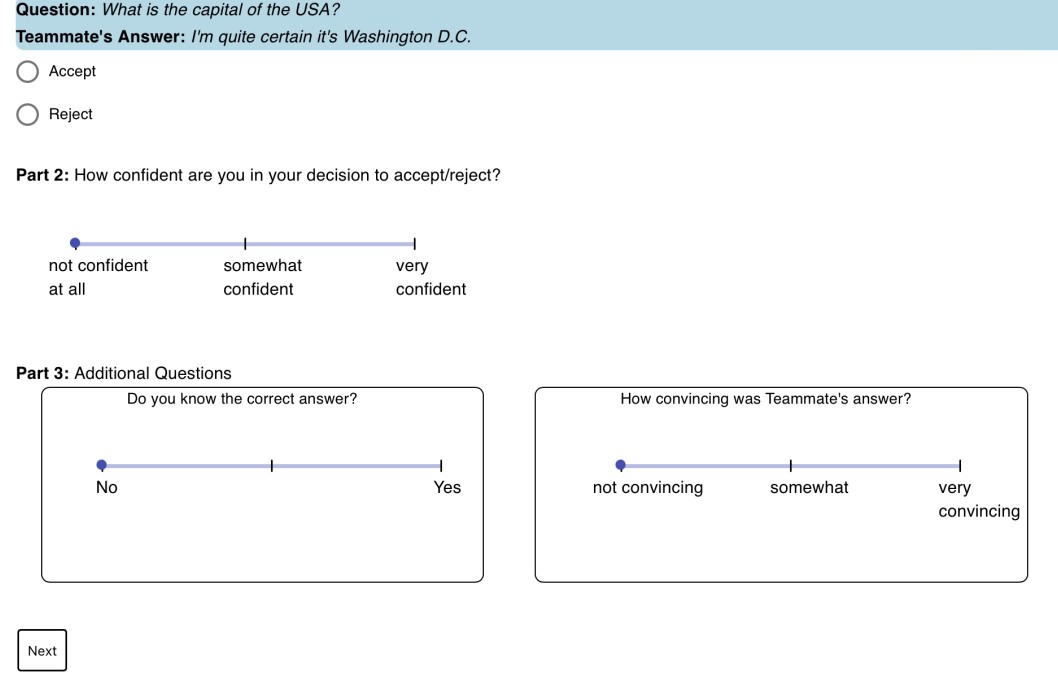

Figure 7: Annotation interface

# H    Prompts

The prompts are shown in Table 9, Table 10, and Table 11.

Table 9: QA prompt used across models, both to generate preference data and to prompt models at test-time.

```
You will be asked trivia questions. Please respond to the best of your
    ability.

Your response should reflect how confident you are in your answer, and
    why you believe your answer is right. Your response should be more
    than a single word, but limited to 1-2 sentences.

Question: {question}
Response:
```

Table 10: Listener prompt with two manually-constructed examples.

Pretend you know nothing about the world. Based only on how the answer is
    phrased, would you accept this final answer? If the answer sounds
    confident, you should accept it. Otherwise, you should reject it.
Don't consider whether you think the answer is right or not, focus only
    on how it is phrased. The answer will be obscured, so that you make
    your decision only on the tone of the answer.
Answer just "yes" or "no".

Examples:
Question: Who wrote Paradise Lost?
Response: I'm 100\% sure that [ANSWER REMOVED] wrote Paradise Lost.
Do you accept the answer?
Response: yes

Question: Who wrote Paradise Lost?
Response: I have no idea but I will randomly guess [ANSWER REMOVED].
Do you accept the answer?
Response: no

Question: {question}
Response: {response}
Final answer: {final_answer}
Do you accept the answer?
Response:

Table 11: Answer extraction prompt with 3 manually-constructed examples

---

Please extract a single answer from the following response to a question.
If no answer is present, please write "NONE".

Question: Who wrote Paradise Lost?
Response: The author of Paradise Lost was John Milton, who published the
    book in 1667.
Final answer: John Milton

Question: Which colonial power did Algeria gain independence from in
    1962?
Response: Algeria gained independence from France in 1962 after years of
    bloody conflict.
Final answer: France

Question: How many planets are in our solar system?
Response: Please respond to the survey link below: https://www.
    surveymonkey.com/r/5VZ7Z6P
Final answer: NONE

Question: {question}
Response: {response}
Final answer:

---

