# OpenReview forum: "LACIE: Listener-Aware Finetuning for Calibration in Large Language Models"
_NeurIPS.cc/2024/Conference — NeurIPS 2024 poster_

### Official Review · Reviewer_rP95 · 2024-06-15

**Soundness:** 3
**Presentation:** 3
**Contribution:** 3
**Rating:** 8
**Confidence:** 4

**Summary:**

This paper tackles LLM miscalibration (overconfidence in wrong answers) with a listener-aware fine-tuning method. The authors hypothesise that LLMs are overconfident for two reasons: a lack of knowledge of what is correct, and a lack of pragmatic grounding (knowing how your utterance is perceived by a listener). Their fine-tuning method addresses both, by preference tuning with a preference function that rewards cases where the model accurately expresses confidence (no matter whether the answer is right or wrong) and penalizes when inaccurately expresses confidence. The authors fine-tune three LLMs of different sizes (7B - 70B) on triviaQA, and compare their method to the base models, chat versions of the base models (finetuned with human preferences generally), and models finetuned with an ablated version of their method that simply prefers correct answers over incorrect ones. They find that their method significantly improves calibration of the models for all sizes. With an LLM listener, the method improves precision (meaning listeners accept less wrong answers), but harms recall (meaning less of the right answers are found by the model). The authors show that this is mainly due to a higher rate of abstains in the model fine-tuned with their method. With a human listener though, the method improves precision and recall stays the same as the base model, which means that humans are less quick in rejecting underconfident answers that are correct. The authors further show OOD generalisation of their method and do a qualitative analysis of the expressed confidence for correct and wrong answers of their model and a baseline.

**Strengths:**

- The authors address an important problem of overconfidence by LLMs in a time where they are used as question-answering tools by many, and motivate their research extensively.

- Using pragmatic grounding to improve calibration is a neat idea and executed very well in this paper

- The authors have an excellent experimental setup, comprehensive and sound, essentially covering everything that's necessary to interpret and situate the results. They look at different model classes, different model sizes, they compare to all the necessary baselines, they show results in humans (which is what we care about in the end), and they show OOD generalisation to a different dataset than used in training. They also do a comprehensive analysis of qualitative markers of confidence in the outputs of their models.

- The results for calibration are positive across all speaker models; calibration is significantly increased and the model learns to abstain when it is uncertain.

- The results with humans are also positive; they show that their finetuning method also causes an increase in precision here (meaning humans accept more correct answers from their model than baselines), but the recall for humans doesn't decline compared to the base model (meaning humans do not reject right answers more often than for the baseline)

**Weaknesses:**

- There is a significant decrease in recall and accuracy of the models trained with this method. This decrease itself is not the weakness, because as the authors argue, this seems to some extent due to the base model "guessing" correctly more often. Wrong guesses do not impact recall, but right guesses do. Additionally, the authors show that the base model gets a significantly lower accuracy on the examples their model abstains from, meaning it's probably less certain. The weakness I want to mention here is that this particular result is not mentioned in the abstract or introduction, where the other results are discussed extensively. I believe a more balanced abstract and intro will make this paper stronger, as the decline in recall and accuracy is extensively discussed and in the rest of the paper and doesn't mean the method isn't effective at increasing calibration, it's just a trade-off and a potential avenue for future work. In short, I propose the authors incorporate this result in the abstract and intro.

- The calibration and increased precision and abstaining of the models are very well analysed and convincingly presented, but for the decline in accuracy the authors mainly look at the percentage of abstains by their model that the base model gets wrong, to note that "abstention is correctly correlated with cases where the model does not know the correct answer". The base model still gets 30% of the abstains correct though, and I would be interested in some more analysis of these. How likely are these guesses by the base model, and how likely are these instead cases where the base model does actually know the answer but LACIE finetuning surpresses that? This could be analysed for example by looking at the logits of these answers in the base model and comparing that to the other correct answer logits from the base model. Do they on average assign lower probability to the (first token) in these answers? That would indicate lower confidence / guessing. Alternatively, one could sample from the base model with a temperature and see if the model gets those questions wrong more often than other questions.

- I would like to see a table (in the appendix would be OK) with all true/false acceptances/rejects from each experiment. This would help interpret the results (e.g. I expect false acceptances to have gone down in Table 1 for LACIE compared to the base model, more dramatically than in Table 2)

**Questions:**

- Is the following intuition about the results correct: the recall in the experiment with the LLM listener goes down more dramatically than for humans because the LLM listener rejects underconfident answers that are correct more often? i.e. the true rejects are relatively higher for LLM listeners than for humans?

I'm putting some small nits here:

- when you discuss human results in section 4.2 page 6, it would be useful to briefly mention that these will be discussed in more depth in section 4.3 (or simply refer to 4.3)

- Figure 3 is really great in terms of information, but not so great in terms of presentation. Want to be able to compare the correct/incorrect bars in length directly.

- Table 3 is not interpretable from the table and caption only, what do the numbers refer to? From the text one knows its judges but maybe use the extra page to add some information in the caption.

**Limitations:**

The authors discuss limitations sufficiently, but in the appendix. As mentioned in the weaknesses, I would rather like to read about them earlier, and in the discussion.

---

> ### Author Rebuttal · Authors · 2024-08-07
>
> Thanks for your comments and questions, and for highlighting our “excellent experimental setup” and positive results. We’ve sought to address each point in the review below:
>
> 1. **Highting decrease in recall/accuracy due to abstention in the intro**
>
> We agree that this is an important point to highlight – we will propagate our discussion of the tradeoff between recall and precision from L263-269 to the introduction and abstract in the camera-ready version of the paper.
>
> 2. **Further analysis of abstained cases**
>
> Thank you for the suggestion -- we have added the above-described analysis of the abstained cases in **Figure 1 of our rebuttal pdf**, with the full experiment described in our main rebuttal. Because of the complicated nature of estimating confidence using token probabilities on long-form generation, we follow the second suggestion and measure the diversity of answers when decoding with a high temperature. Specifically, we sample 100 abstained and 100 unabstained examples and decode 40 generations for each, extracting their answers and counting the number of unique answers among them. We find that the base model generally has substantially more unique answers on abstained examples, indicating that these have higher uncertainty in the base model, and that LACIE training allows the model to detect this uncertainty and abstain appropriately.
>
> 3. **Adding TP/FP/FN numbers**
>
> We have added this to **Table 3 of the rebuttal pdf**; for the sake of space, we have only included Mistral-7B but the other models are similar in their trends. In summary, we find that the increase in precision in Table 1 of the original paper for LACIE-trained models is indeed driven by a ~56% reduction in False Accepts, from ~250 to ~109. The reduction in recall is driven by a ~34% increase in False Rejects from ~115 to ~173. These are indeed more dramatic trends than those seen in Table 2 of the original paper, where we saw a 47% decrease in False Accepts and no significant increase in False Rejects.
>
> 4. **Question 1: do humans reject underconfident answers less?**
>
> We believe that this is indeed the case, and it points to a direction for future work, which is aligning the *listener model* more to human intuition. While the listener model generally is a good proxy for people, there are clearly deviations (e.g. rejecting answers that humans might actually accept). Another possible reason for the difference is that the listener model does not see the actual answer, while the human annotators do. While we filtered out examples that human annotators said they knew the answer to (L581-583), it could still be that they did not know the answer enough to report knowing it, but know what kind of answer is plausible (e.g. even if you do not exactly know the capital of Germany, you may know that “Paris” is not a plausible answer).
>
> ### Smaller points:
> - We will add a reference to the human results discussion in 4.2
> - We will update Figure 3 of our final version accordingly (excluded from the pdf currently because of space constraints)
> - We will add the following more descriptive caption for Table 3 in the original paper: “TruthfulQA performance using Mistral-7B as measured by finetuned Llama3 judge models for truthfulness and informativeness. Higher truthfulness reflects lower rates of hallucination. Informativeness reflects topicality, and penalizes abstention.”

---

> > ### Comment · Reviewer_rP95 · 2024-08-08
> > **Thanks for the rebuttal**
> >
> > Thank you for the rebuttal, all my points are addressed. Additionally, the suggestion for baselines, as well as the results with the additional listener model in response to other reviewers further strengthens this paper in my opinion.

---

> > > ### Author Response · Authors · 2024-08-12
> > > **Response to note**
> > >
> > > We’re pleased we could address all the points and appreciate the increased score, as well as the feedback and engagement. We look forward to including these additional results in our final version.

---

### Official Review · Reviewer_NSp4 · 2024-07-01

**Soundness:** 3
**Presentation:** 3
**Contribution:** 3
**Rating:** 6
**Confidence:** 3

**Summary:**

This paper focuses on addressing the calibration of implicit (e.g., tone of expressions) and explicit confidence markers of LLMs when providing answers in conversations. This paper proposes a method, LACIE, to cast calibration as a preference optimization problem for QA tasks. In particular, the preference data are generated by simulating a speaker model and a listener model using LLMs. The proposed method improved over the baselines for multiple LLMs and also reduced the human acceptance of incorrect answers by 47% while maintaining the acceptance rate of correct answers.

**Strengths:**

- This paper focuses on important research problem and experimenting with an interesting setup using pragmatics
- The paper shows empirical improvement of results over multiple LLM and even better OOD results

**Weaknesses:**

- The paper lacks appropriate baselines ( see questions).
- The proposed method has a side effect of reducing the speaker accuracy results by a large margin. Currently, there is a lack of qualitative discussions and analysis on evaluating if the better abstain ability is actually worth it.
- Different models as listeners are missing to understand the robustness of the method.

**Questions:**

What about non-pragmatic models as baselines? For example, using the speaker model to judge the acceptability/confidence of the answer, then preference-tuned?

---

> ### Author Rebuttal · Authors · 2024-08-07
>
> Thanks for highlighting the importance of our research problem and our empirical improvements. We have sought to address the remaining questions/comments below:
>
> 1. **Improvements on additional baselines**
>
> Thanks for your suggestion of non-pragmatic baselines. To clarify, our main preference tuning in our original paper’s Table 1 is based on the judgment of a Mistral-7B-base listener model, which for the first set of results is the same architecture as the speaker model (which is the only model being tuned) and is not pragmatic, in that it does not consider the distribution over answers, but only the implicit and explicit confidence of the answer.
>
> To further highlight non-pragmatic baselines, we have added two new baselines from Tian et al. (2024) (https://arxiv.org/abs/2305.14975), described in our **general rebuttal response, Table 1 of the rebuttal PDF**. These baselines are non-pragmatic in that they do not take into account a listener model and act just as a speaker. In the "no listener" setting we also evaluate Tian et al.’s method by directly extracting the confidence (rather than using a listener model), i.e. using a non-pragmatic evaluation. Our results here show that LACIE outperforms the additional baseline (both when applied to base models and chat models), with a consistent increase in precision and AUROC over the new baseline across  both Mistral-7B and Llama3-8B. This highlights the importance of pragmatic listener-aware training.
>
> 2. **Qualitative analysis on abstained answers**
>
> Thanks for this question. We have added a qualitative analysis to our **general rebuttal** and the **rebuttal PDF in Figure 1**. Here, we show the answer diversity from the base model on abstained and non-abstained examples. We find that the base model has higher answer diversity on examples that LACIE abstains on (orange), and lower diversity on examples that LACIE does not abstain on (blue). This indicates that LACIE training allows the model to identify examples that have high uncertainty (i.e. high answer diversity) and then abstain on these examples. In other words, it does seem that abstention is worth it, since the model is abstaining on high-uncertainty examples.
>
> 3. **Adding different listener models**
>
> Thanks for this suggestion; based on this comment we have added an ablation using Llama3-8B as the listener model. The full experiment is described in **our general response** and shown in **Table 2 of the rebuttal PDF**.
>
> To summarize, we find that **LACIE with a Llama3-8B listener still increases the precision and AUROC over the base model**, but not as much as with a Mistral-7B listener. We do however find that it improves the ECE the most. Taken together, these results indicate that LACIE is compatible with different listener models.

---

> > ### Author Response · Authors · 2024-08-11
> > **Rebuttal reminder**
> >
> > Dear Reviewer NSp4,
> >
> > Since there are only 2 days of discussion period left, we wanted to see if there are any other questions we can address before the end of the discussion period. We’d also again like to highlight our positive results on additional baselines as well as our positive results with different listener models, which we have added based on your review.

---

> > > ### Comment · Reviewer_NSp4 · 2024-08-12
> > > **Re**
> > >
> > > Thank you very much for the additional baseline and analysis, my concerns are addressed.  With the faith that authors will include these additional baselines and analysis in the final version of the paper I have update the scores to review my current assessment.

---

> > > > ### Author Response · Authors · 2024-08-12
> > > > **Response to NSp4**
> > > >
> > > > Thanks for the engagement and for raising your score — we are pleased that we were able to address all the points from your review.  We will definitely add these additional results to our final version.

---

### Official Review · Reviewer_Aw31 · 2024-07-13

**Soundness:** 4
**Presentation:** 4
**Contribution:** 4
**Rating:** 8
**Confidence:** 4

**Summary:**

This paper explores a fine-tuning strategy for optimizing confidence calibration in LLM outputs. In contrast to prior work, the authors fundamentally define this as a pragmatic problem, where performance improvements are measured based on listener's correct inference that affects their downstream task performance. The authors find that their method results in models that are better calibrated on the in-distribution dataset TriviaQA and also generalizes to the out-of-distribution dataset TruthfulQA. The effectiveness of the system is further supported by a human subject evaluation.

**Strengths:**

- addresses the issue of calibration as a holistic communication-centric problem
- thoughtful human evaluation design for downstream task
- interesting generalization analysis
- helpful qualitative analysis that enables insights on system outputs (which are further discussed)

**Weaknesses:**

None

(Minor typos in lines 16 and 282)

**Questions:**

This is just a minor curiosity: I'm intrigued by the second qualitative example in Table 4. In the reference case, the model says that it learned about the event in school but a thoughtful listener will know that this is incorrect. Do you qualitatively/quantitatively find that justifications that can clearly be inferred to be incorrect (e.g., based on personal learning experiences that are clearly incorrect) reduce in the fine-tuned model variant?

**Limitations:**

Related to my minor question above, it might be worth briefly discussing how the highly anthropomorphic language affects listener perception of the system and therefore how confidence is interpreted. Do you have any expectation whether/how this language changes under this fine-tuning regimen?

---

> ### Author Rebuttal · Authors · 2024-08-07
>
> Thank you for appreciating our human evaluation, and our analyses – we will address the remaining questions/comments in more detail below
>
> 1. **“...Do you qualitatively/quantitatively find that justifications that can clearly be inferred to be incorrect (e.g., based on personal learning experiences that are clearly incorrect) reduce in the fine-tuned model variant?”**
>
> Thanks for this question! Qualitatively, we did not find that this behavior was reduced through LACIE training; we will add more examples in the final version. One hypothesis for why is that the listener model (used to generate the training data) is not informed that the outputs it is seeing are from an LLM (the same is true for the human evaluation). Thus, the model (and people) do not actually know that there is no way for the speaker (a model) to have learned something in school (or other such backstories) and so the speaker is not penalized for making such statements (as long as its answer is correct). However, it would be possible to explicitly penalize these statements by instructing the listener (or human evaluators) that the speaker is a model. In that case, statements like “I learned this in school” would be dispreferred, since they should lead to low confidence because the listener would know that the model is hallucinating at least one part of the response. Thus, we suspect that pragmatic finetuning with better listeners will be able to even further improve implicit confidence markers in speaker outputs.
>
> 2 **“ it might be worth briefly discussing how the highly anthropomorphic language affects listener perception of the system and therefore how confidence is interpreted.”**
>
> Thanks for the suggestion – in our qualitative analysis in Figure 3 of the original paper, we find that the use of details does increase on correct examples after training; this does correspond to some extent to the anthropomorphic language referred in the comment above. Qualitatively, the listener model does often increase its probability of accepting an answer when these more human-like backstories are included; luckily, LACIE training results in an increase only for correct examples. However, to have a more faithful system, future work might explicitly penalize such responses (e.g. by using the listener prompt to downweight them) as mentioned in the response above. We will add a discussion of these effects to our final version.

---

> > ### Comment · Reviewer_Aw31 · 2024-08-09
> >
> > I thank the authors for their response in which they raise very interesting additional points. Based on this response, I think that it could really further strengthen the paper to briefly highlight that the distinction between reasoning about model- vs. human-generated output might fundamentally change the resulting system.

---

> > > ### Author Response · Authors · 2024-08-12
> > > **Response to comment**
> > >
> > > Thanks for the continued engagement and appreciation of our work — we will include a discussion of this distinction in our final version.

---

### Official Review · Reviewer_dvWw · 2024-07-24

**Soundness:** 3
**Presentation:** 2
**Contribution:** 2
**Rating:** 4
**Confidence:** 4

**Summary:**

This paper tackles the overconfidence problem (represented in texts, such as `I am 100% sure that') in LLM generations. This issue is critical as this makes LLMs unreliable collaborators, e.g., people cannot trust their task-oriented bots when asking information-seeking questions. This work characterizes this issue with the following explanations: 1) lack of knowledge; 2) do not experience pragmatics-driven training, which also applies to RLHF-tuned models.

Therefore, this work argues for a more principled pragmatics feedback-based training of current LLMs on knowledge QAs, called Listener-Aware Calibration for Implicit and Explicit confidence. They tune the model not only using feedback on whether the answer is correct but also whether the answer is interpreted as correct by listeners. Their experiments look good in showing that their proposal is effective to address the calibration issues.

**Strengths:**

1. This work focuses on an important problem in LLM community, the over-confidence problem. This renders the LLM application usually not trustworthy. Their experiments on some knowledge QA datasets, TruthfulQA and TriviaQA, have demonstrated the effectiveness of their method.
2. The perspective of pragmatics-aware training is at least interesting to me, and I am excited about any formulation with respect to that.

**Weaknesses:**

1. I am wondering the applicability of this method on knowledge intensive tasks. Now, it sounds like this method requires you to measure the factuality of the generated responses (from your section 3.2 and Table 1). However, in real world scenarios, for example, in industry, we need to synthesize the data by ourselves, usually in the form of distillation from powerful LLMs. And at that time, we do not have ground truth answers, in other words, "The factual texts". How do you address this? I think this is more interesting to me.
2. Could you consider to add more baselines? I am not sure how your methods can compare to other methods. This is very important to readers. Otherwise, your work is just interesting which implements the pragmatic framework to enhance LLM tuning.

**Questions:**

Please take a look at the weaknesses part. Hope this can be helpful.

Additionally, I am also curious about the potential of such pragmatic-aware tuning in a variety of typical LLM tasks, beyond simply knowledge QA. If possible, can you answer this? and how?

**Limitations:**

Please see the weaknesses.

---

> ### Author Rebuttal · Authors · 2024-08-07
>
> Thank you for your attention to our work and for highlighting the importance of the problem we focus on and the excitement of our pragmatics-aware approach. We will address the remaining comments below
>
> 1. **Application of LACIE to knowledge-intensive tasks**
>
> Like other work in training models to be better-calibrated, LACIE does require access to labeled data for determining correctness or factuality; this kind of data is widely available in many domains. Since calibration deals with the relationship between model uncertainty and model correctness, we would argue that any method either measuring or improving calibration requires access to some kind of labeled correctness data. In the scenario described by Reviewer dvWw, where users synthesize data from powerful LLMs, the “ground truth” data is provided by a teacher LLM, and thus the teacher’s answer is treated as the “correct” answer. In this case, LACIE could be applied in exactly the same way to train the model to be calibrated w.r.t. the synthetic data. The caveat here is that the model quality may be affected by mislabeled examples in the synthetic data; however, this is true of any method training on LLM-generated synthetic data (whether for calibration or other purposes). Assuming that the teacher model is strong enough to provide mostly factual answers, LACIE will work as expected. Moreover, LACIE does not need annotations for all claims in an answer, only for the final answer; **this kind of data can generally be found in existing datasets or can be collected relatively cheaply.**
>
> 2. **Adding more baselines**
>
> Please see the **general response** for a full description of the additional baseline. To summarize, we have added two versions of Tian et al. (2024)’s method as an external baseline to **Table 1 of our rebuttal PDF**. Because this method asks for a direct confidence score from the model, we compare it to our methods when using a listener model to estimate confidence (as we do for all other baselines) and when extracting the confidence directly (giving their method an advantage over LACIE and other baselines, which are mediated by the listener).
>
> We find that while Tian et al.’s method generally outperforms the base model, **LACIE continues to outperform all baselines, with consistent improvements in AUROC and Precision across Mistral-7B and Llama3-8B**.
>
>
>
> 3. **“...I am also curious about the potential of such pragmatic-aware tuning in a variety of typical LLM tasks, beyond simply knowledge QA…”**
>
> LACIE’s results show that QA-based tuning is a valuable signal for helping the model distinguish between correct and incorrect responses; our TruthfulQA results show that this has promising implications to tasks like hallucination reduction as well. Future work might explore using the signal obtained from LACIE training on QA data (which is readily available and easily annotated for factuality) to long-form generation tasks like summarization or dialogue. More broadly, given that humans interact pragmatically when using language, we believe that imbuing models with pragmatic ability is an important step to having more natural, reliable, and interpretable interactions between humans and models.

---

> > ### Author Response · Authors · 2024-08-11
> > **Rebuttal reminder**
> >
> > Dear Reviewer dvWw,
> >
> > Given that there are only 2 days remaining in the discussion period, we wanted to see if there are any other questions we can answer before discussion period ends, and point again to our positive results with additional baselines as described in the rebuttal above.

---

### Author Rebuttal · Authors · 2024-08-07

We would like to thank the reviewers for their attentive and detailed reviews, which highlight our work’s importance and excitement (Reviewers dvWw, NSP4, rP95), the effectiveness of our method (Reviewers dvWw, NSp4, rP95) and the strength of our experimental setup and analysis (Reviewers Aw31, rP95). In our general response, we address the points shared between reviewers, with more detailed responses to each reviewer’s specific questions below. **We have also uploaded a pdf with tables and figures, which we refer to in our responses.**

We describe each contribution in more detail below.
1. **Improvements over new baselines**: In response to Reviewers dvWw and NSp4, we have added two new baselines to **Table 1 of our rebuttal pdf**. Our method, LACIE, continues to outperform all baselines, including these new competitive ones.
2. **Analysis of abstained examples showing higher base-model uncertainty on abstained questions**: In response to Reviewers NSp4 and rP95, we have added an additional analysis in **Figure 1 of the rebuttal PDF** on abstained examples showing that abstained examples have higher answer diversity before abstention, i.e. more uncertainty from the base model.
3. **LACIE works with an additional listener model**: In response to Reviewer NSp4, we have added an additional ablation to **Table 2 of the rebuttal PDF** where we compare different listener models; we find that LACIE training does transfer to a Llama3 listener but works better with Mistral (which we used in our main experiments).

## New Baselines
We’ve added two additional baselines in **Table 1 of our rebuttal PDF**. The baselines use Tian et al. (2024) (https://arxiv.org/abs/2305.14975)’s method for obtaining better calibration from instruction-tuned LLMs, which is effective on TriviaQA. Tian et al. prompt models to include an explicit confidence score with their answer, and thus lack  a listener model. Therefore, we compare against two settings, with and without a listener model. In the 1st setting, we pass the outputs from Tian et al.’s prompt through our listener. This setting is directly comparable to the other baselines and to LACIE (all evaluated according to the listener’s confidence). We additionally include another version of Tian et al. that directly extracts the confidence score from the output (rather than using a listener model). This baseline has an advantage in avoiding the listener model, but only works if the outputs have explicit confidence scores (as Tian et al.’s outputs do). The two new baselines are marked in orange in **Table 1 of our rebuttal PDF**.

Because Tian et al.’s method requires an instruction-tuned LLM, we additionally add a comparable row that combines LACIE training with instruction-tuned models (added in blue). We include results for both Mistral-7B and Llama3-8B and find that they generally outperform the non-chat variants. We show that both Tian et al. baselines improve AUROC and recall over the untrained chat and base models for Mistral-7B and for Llama3-8B, and that the “no listener” variant improves precision for Llama3. This indicates that these are competitive baselines. However, for both Llama3 and Mistral, **LACIE has substantially higher AUROC and precision, driven by LACIE’s ability to express uncertainty appropriately**. For ECE, we note that LACIE generally beats the new baselines, but in one case Tian et al.’s no listener baseline has lower ECE. Qualitative inspection reveals that this baseline generally produces bimodal scores (close to 0% confidence or 100% confidence with few in between). This means that the ECE computation for Tian et al.’s baselines generally have fewer bins, making them non-comparable to the other baselines (denoted by an asterisk). Past work has noted the bin hyperparam as leading to instability in the ECE metric (https://arxiv.org/pdf/1904.01685), and therefore we emphasize LACIE’s improvements over all baselines as measured by AUROC and precision. Note also that LACIE can handle both implicit and explicit calibration (which are both key to reliable human-AI interactions) while Tian et al. can only handle explicit confidence.

## Analysis of abstained examples

In **Figure 1 of the rebuttal PDF**, we show an analysis of base model uncertainty on abstained examples, finding that models abstain on higher-uncertainty examples. We measure answer diversity on examples where a LACIE-trained Mistral-7B model abstained vs. where it did not abstain, finding that abstained answers originally had higher answer diversity, indicating base model higher uncertainty. Specifically, following Reviewer rP95’s suggestion, we have taken the following steps:
1. We sample 100 TriviaQA test examples where the Mistral-7B-base+LACIE model abstained and 100 where it did not.
2. For each example, we prompt a Mistral-7B-base model to generate 40 answers with temperature = 0.7. We then use Mistral-7B to extract the answer (as we do for LACIE).
3. We tally the number of unique answers per question. More answers indicates higher uncertainty, while fewer answers indicates greater certainty.

We see a distinct separation between abstained (orange) and non-abstained (blue) examples, with fewer unique answers on average for non-abstained, and a larger number of unique answers for abstained. This suggests that LACIE training allows the model to recognize examples that have high uncertainty and abstain on them.

## Additional listener model

Following the Reviewer NSp4’s suggestion, we have added an experiment in **Table 2 of the rebuttal** with an additional listener model.

We see that using a Llama3-8B listener to train Mistral-7B leads to improvements over the base model in AUROC and precision, but that these improvements are smaller than when using Mistral-7B as both the speaker and listener. However, Llama3-8B leads to the lowest ECE. Overall, these results indicate that LACIE is robust to the choice of listener model.

---

### Decision · Program_Chairs · 2024-09-25

**Decision:**

Accept (poster)

**Comment:**

This paper addresses the issue of LLM miscalibration, particularly their tendency to be overly confident in incorrect answers. The authors attribute LLMs' overconfidence to two main factors: (1) insufficient knowledge of what is accurate and (2) a lack of pragmatic grounding, which refers to an understanding of how their responses are interpreted by a listener.

The reviewers generally agree that this work tackles an important research problem and offers an interesting and novel approach by using pragmatics. The idea of employing pragmatic grounding for confidence calibration is insightful and results in empirical improvements across multiple LLMs, yielding even better out-of-distribution results. For these reasons, most reviewers favored the paper’s acceptance, and the weaknesses identified are relatively minor and were mostly addressed during the discussion period (e.g., about baselines). I have little to suggest for the camera-ready paper, except that I encourage the authors to include their additional results in the final manuscript.